# Barriers to the Utilization of Low-Vision Rehabilitation Services among Over-50-Year-Old People in East and Southeast Asian Regions: A Scoping Review

**DOI:** 10.3390/ijerph20237141

**Published:** 2023-12-04

**Authors:** Saito Takashi, Imahashi Kumiko

**Affiliations:** Department of Social Rehabilitation, Research Institute of National Rehabilitation Center for Persons with Disabilities, 4-1 Namiki, Tokorozawa 359-8555, Japan; imahashi-kumiko@rehab.go.jp

**Keywords:** East Asia, Southeast Asia, low-vision rehabilitation, accessibility, barrier, scoping review

## Abstract

East and Southeast Asia (ESEA) are facing age-related eye health issues. Low-vision rehabilitation (LVR), which is a special rehabilitation service for individuals with vision impairment, is a promising solution for these health issues; however, poor accessibility to LVR services has been reported globally, including ESEA. Therefore, this scoping review aimed to summarize and understand the barriers to accessing LVR services in ESEA. In total, 20 articles were ultimately considered eligible for this scoping review after an electronic database search using MEDLINE (PubMed), Web of Science, Academic Search Ultimate (EBSCO), and Ichushi-Web (Japanese medical literature database), and an independent review by two reviewers. Twenty-one potential barrier factors were identified in the full-text review. Notably, age, education, economic status, “previous experience using eye care service”, and “knowledge, information, and awareness” were the possible barrier factors that were examined for their association with LVR utilization, with supportive evidence in many eligible studies. We also identified research gaps relating to geographical and ethnic diversity, the scope of LVR services, and barriers among eligible articles. Therefore, by conducting further studies addressing the research gaps identified in this scoping review, these findings can be used to make LVR services more accessible to people in ESEA.

## 1. Introduction

East and Southeast Asia (ESEA), with an estimated more than two billion people spread across ethnically, economically, and politically diverse countries, is a region facing an aging population crisis [1]. According to a report published by the United Nations World Population Aging 2019, ESEA was home to the largest number of older persons as of 2019 (261 million), with a projected increase in the population from 261 million in 2019 to 573 million in 2050 [2]. Population aging can create various challenges associated with aging-related health conditions for both individuals and society as a whole, such as the rising demand and expenditure for health services, growing requirements for long-term care, and increasing needs for income and social security [3]. Therefore, one of the prioritized challenges for the ESEA society is establishing efficient and effective healthcare systems for managing and mitigating the impact of the burden associated with aging-related health conditions [1].

In the context of the aging population in ESEA, one crucial area that necessitates a good healthcare system is eye care health. Aging has been well-documented as one of the leading contributing factors to vision impairment [4]. The risk of aging-related eye conditions such as cataracts, presbyopia, glaucoma, and age-related macular degeneration can increase with age [5]. For instance, presbyopia (farsightedness) has affected more than 500 million people in the ESEA and Oceania regions, which is the largest number of people globally [5]. Moreover, it is noteworthy that approximately half of these conditions remain unaddressed in ESEA [5]. This implies that access to eye care health services is hindered by certain barriers, resulting in inaccessible eye care health services and inequality in eye health status in the ESEA regions. Therefore, strategies to overcome barriers to service utilization should be incorporated into the eye healthcare system of ESEA.

Low-vision rehabilitation (LVR) is a special rehabilitation service for individuals with vision impairment which can help in improving their eye health status. The LVR service, which originated in Western countries in the 1950s, aims to reduce the impact of visual impairment and minimize disability using one or more concurrent approaches, such as the prescription of devices, training in their use, and adaptation to the environment [6]. Systematic review articles have shown that LVR can improve the activities of daily living and quality of life in individuals with vision impairment [7,8].

Despite previous studies showing LVR services to be a promising eye care service, poor accessibility to LVR services has also been reported in high-, low-, and middle-income countries [9]. Although the LVR service holds promise for people with vision impairment, inaccessibility can compromise the potential of LVR to improve eye health status and the well-being of people in ESEA. Therefore, a better understanding of the barriers to LVR utilization is required. This knowledge can enable policymakers and eye care professionals to make informed decisions when establishing and operating good eye health systems that can provide LVR services equally for all individuals in need. Understanding the locally contextualized evidence in the ESEA regions is crucial for establishing a locally applicable and practical eye health system. Several existing articles [10,11,12] have reviewed and summarized the available information on barrier factors to LVR utilization. However, these studies did not provide locally contextualized information on the barriers to LVR utilization in the ESEA regions.

Therefore, this scoping review aimed to summarize and understand the available evidence on LVR service utilization in ESEA. Specifically, we focused on scientific findings regarding the barrier factors that hinder LVR service utilization among middle-aged and senior populations in ESEA. We defined the middle-aged and senior populations as individuals aged > 50 years based on scientific estimations that the burden of vision impairment was greatest in those aged ≥ 50 years [4]. Our review question was formulated as what are the barriers to LVR service utilization among people aged > 50 years in the ESEA regions? We believe that locally contextualized evidence from the ESEA regions can contribute to establishing a robust eye health system that can provide LVR services equally for all individuals in need.

## 2. Materials and Methods

### 2.1. Protocol Registration

The protocol for this scoping review has been published [13], titled “Barriers and enablers of the utilization of LVR services among over-50-year-old people in East and Southeast Asian regions: a scoping review protocol”. According to this title, we focused on barriers and enablers of LVR service utilization. However, barriers and enablers are practically two sides of the same coin, which might confuse readers when using both terms in one study. Therefore, to enable this scoping review to provide more straightforward information and evidence, we modified the protocol to focus only on the barriers to LVR utilization.

This article is reported in accordance with the Preferred Reporting Items for Systematic Reviews and Meta-Analyses extension for Scoping Review (PRISMA-ScR) checklist [14].

### 2.2. Definition

East and Southeast Asian countries and regions: these include Borneo, Brunei, Cambodia, China, Hong Kong, Indonesia, Japan, Korea, Laos, Macau, Malaysia, Mongolia, Myanmar, Philippines, Singapore, Taiwan, Thailand, Tibet, Timor-Leste, and Vietnam as defined by the electronic literature search database, MEDLINE (PubMed) MeSH terms.

Low-vision rehabilitation: Any intervention that aims to mitigate the impact of an eye-related health condition-induced disability. Specific LVR services included, but were not limited, to visual function assessment, prescription and/or use of devices (such as spectacles, assistive devices, guide dogs, and canes) and training in their use, adaptations to the environment, interventions to improve orientation and mobility, education for braille or technology use, and providing social and psychological support [6,15]. However, we excluded curative interventions (such as cataract surgery), pharmacotherapy, and preventive activities that focused only on screening individuals at risk of eye diseases or conditions (such as glaucoma or diabetic retinopathy).

Utilization of LVR services: Receiving or taking up LVR services. Additionally, being referred to LVR services was considered one part of the LVR service utilization.

### 2.3. Eligibility Criteria

These were created based on the Population, Intervention, Comparison, and Outcome framework.

Population: The target population comprised individuals with vision impairment aged ≥ 50 years. No restrictions were placed on the participants’ physical or medical characteristics (e.g., sex, race, and type of eye disease) other than their age.Intervention: LVR services.Study setting: ESEA countries and regions.Comparator: not applicable.Outcomes: These included any barrier factors that hinder the LVR service utilization.Study design: We included all types of original peer-reviewed and academic-quality articles aimed at exploring the barriers to LVR service utilization, except for case reports, protocol papers, editorials, and conference abstracts. In these articles, outcome variables were typically set as situations where the LVR service utilization was hampered (e.g., no, or less receiving, uptake, or being referred to LVR services). The relationships between the outcome variables and explanatory (any barrier factors) variables were examined using quantitative, qualitative, or descriptive analysis. These articles, based on a certain analysis, provide information on whether the explanatory variables are barrier or not-barrier factors. The barrier factors mean that the factors are relevant to “less or no LVR service utilization”. While the non-barrier factors mean that the factors are not relevant to “less or no LVR service utilization”. Notably, non-barrier factors do not necessarily mean that the factors are “enablers” for LVR service utilization.Language: due to linguistic barriers, our search included only articles written in English or Japanese.Publication date: The World Health Organization launched VISION 2020 [16] in 1999. This was a significant initiative for the development of the eye healthcare system. However, findings from articles published before 2000 may be outdated and not applicable to the current situation. Therefore, we set the lower date limit as January 1, 2000, while the upper date limit was the day of the literature search.

### 2.4. Source of Information and Search Strategy

Four electronic databases were used for the literature search as follows: MEDLINE (PubMed), Web of Science, Academic Search Ultimate (EBSCO), and Ichushi-Web (Japanese medical literature database). Details of the search strategy are presented in Appendix A. The literature search was performed on 18 and 19 April 2023.

### 2.5. Study Selection

Two reviewers (ST and IK) independently selected articles that met our eligibility criteria. The selection process was categorized into the following two stages. In the first screening stage, two reviewers reviewed the titles and abstracts of the articles identified through a literature search and evaluated them against the eligibility criteria. Subsequently, the articles were classified into three groups based on the reviewers’ judgments as follows: definitely relevant, possibly relevant, and not relevant. Only articles classified as definitely relevant or possibly relevant were included in the second screening stage. In the second stage, the two reviewers reviewed the full texts of the articles and made a final judgment regarding the eligibility criteria for each article as either eligible or not eligible. Any disagreements between the reviewers during the study selection process were resolved through discussion.

### 2.6. Data Charting and Data Items

Data were charted by a single reviewer (ST), while another reviewer (IK) independently verified the extracted data to ensure accuracy. The data charted from the articles are as follows:General information: first author and year of publication.Study characteristics: study design, study setting (e.g., country, and rural/urban), study methodology and design, types of study population, and number of participants.Eye diseases and conditions being studied.Types of LVR.Findings on barrier factors to LVR service utilization (definitions of no or less utilization of LVR service (i.e., outcome variable), names of barrier factors analyzed (i.e., explanatory variable), statistical methods applied (quantitative, qualitative, or descriptive analysis), and findings of analysis (i.e., barrier or non-barriers)).

### 2.7. Data Synthesis and Data Presentation

The barrier factors were classified into three categories based on a representative model of barriers that influence the utilization of LVR services proposed by Southall and Wittich [17] as follows: individual, healthcare setting, and society categories. Explanations for each category were cited from the previous study by Southall and Wittich [17] as follows:Individual: barriers inherent to individuals, which may include personality, age, sex, financial resources, family and social support, personal knowledge of rehabilitation options, expectations, and priorities.Healthcare setting: barriers inherent in the clinical setting, which may include policies and programs focusing on LVR services, attributes of the ophthalmologist and other staff members (i.e., knowledge of LVR services and motivation to pass along information), characteristics of the consultations (i.e., the time allocated and receptivity to a question-and-answer period), and motivation to refer clients to rehabilitation.Society: barriers inherent to the surrounding community, which may include characteristics of the social, demographic, and cultural communities, governmental policies on accessing specific rehabilitation services or devices, and attitudes of others with respect to using these types of services.

General information and study characteristics were summarized using descriptive analyses (numbers and frequencies).

To provide detailed information for readers, we made minor modifications to an originally planned table format to summarize the information on the barrier factors to LVR service utilization. Specifically, for each barrier factor, we added information regarding the number of articles that indicated whether the factor could be a barrier factor or not. Moreover, the articles were stratified into two categories based on the method of analysis applied (i.e., quantitative analysis and qualitative or descriptive analysis).

## 3. Results

### 3.1. Study Selection

Figure 1 illustrates the study selection process. Initially, 2157 articles were identified through a literature search. After removing 495 duplicate articles, 1662 articles were screened. Of the 1662 articles, 81 were categorized as definitely relevant or possibly relevant. The full text of these 81 articles was reviewed and 18 promising articles [18,19,20,21,22,23,24,25,26,27,28,29,30,31,32,33,34,35] were extracted.

Of the 18 articles, 4 [23,24,30,32] included only insufficient information on participants’ ages and had ambiguity regarding whether the age of study participants met our inclusion criteria or not (≥50 years). Therefore, one reviewer (ST) sent an e-mail to the authors of the four articles to inquire about the participants’ age. Unfortunately, no answers were obtained from the authors of these two studies [23,24]. One study author [30] responded that the requested information was unavailable because the study was conducted more than 5 years ago, and the row data were no longer kept. Another author [32] provided information on the study participants’ mean age (50.5–52.5 years). Regarding the three articles [23,24,30] with insufficient information, the two reviewers (TS and IK) discussed the eligibility and conjectured that the study participants’ mean age would exceed 50 years based on the information of the range and distribution of the study participants’ age. These four articles were finally judged eligible.

Two additional articles [36,37] were extracted through a manual search of the reference lists of 18 eligible articles. Finally, a set of 20 articles [18,19,20,21,22,23,24,25,26,27,28,29,30,31,32,33,34,35,36,37] was included as eligible articles for this scoping review. Detailed information extracted from each eligible article is provided in Appendix A.

### 3.2. General Information and Characteristics of the Eligible Articles

Table 1 summarizes the general information of the eligible articles. Approximately half of the articles included study participants from China (*n* = 7, 35%) and Singapore (*n* = 4, 20%), followed by Cambodia (*n* = 2, 10%), Japan (*n* = 2, 10%), Timor-Leste (*n* = 2, 10%), and three other countries (*n* = 1). Seventeen articles (85%) included results from quantitative study methodology. In total, 10 (50%), eight (40%), and two (10%) articles focused on the general population, individuals with visual impairment, and indigenous or minority people, respectively.

Table 2 presents the characteristics of the 20 eligible articles. More than three-fourths of the eligible articles (17/20 articles) focused on LVR services related to spectacles or contact lenses, with 16 articles including individuals with refractive errors (nearsightedness and/or farsightedness) as study participants. In 11 of the 17 articles, the state of uncorrected and/or under-corrected refractive error was used as the definition of “no or less utilization of LVR services”, while seven articles used open-ended or closed questions for defining “no or less utilization of LVR service” (cf. one study [24] used both the definition of the state of uncorrected refractive errors and the question related to LVR utilization). In addition to spectacle-related LVR services, three other types of LVR services were examined in the eligible articles. These LVR services were as follows: comprehensive vision rehabilitation [32], information on “aids and equipment” and “benefits and money [34]”, and occupational therapy LVR [35].

Of the three categories of barrier factors to LVR service utilization, the most commonly examined was the individual category, with 19 articles examining the relationship between barrier factors in the individual category and LVR service utilization. However, only a few articles focused on the category of healthcare settings (two articles) and society (three articles).

### 3.3. Barrier Factors to Low-Vision Rehabilitation Service Utilization

Table 3 summarizes the barrier factors to LVR service utilization. In total, 21 factors were examined as potential barrier factors to LVR service utilization in the 20 eligible articles. The number of articles that examined the relationship with LVR service utilization varied depending on the barrier factors, ranging from 1 to 13 articles. Seventeen potential barrier factors were from the individual category, while two were each from the healthcare settings and society categories.

#### 3.3.1. Potential Barrier Factors from the Individual Category

Five factors, namely sex, age, education, economic status, and severity or type of eye condition, were the most commonly examined factors, with more than half of the eligible articles examining these factors. Of the five barrier factors, age (older) (8/13, 62%), education (lower) (7/10, 70%), and economic status (less privileged) (8/10, 80%) were reported as the potential barrier factor in approximately two-thirds of the articles focusing on each barrier factor. However, sex (female) was reported as a non-barrier factor in eight of the 12 articles (8/12, 67%). The findings on the severity or type of eye conditions were mixed. Eight (80%) of the 10 articles provided the findings that the factor could be a barrier to LVR service utilization, whereas six (60%) provided the opposite findings. The findings on these potential barrier factors, except for economic status, were mainly based on multivariate analysis after adjusting for potential confounders.

In 13 articles, age was investigated as a potential barrier factor to LVR service utilization. Six studies [19,25,26,29,30,36] found age (older) to be a potential barrier factor based on multivariable analysis and reported a significant increase in the likelihood of no or less utilization of LVR services (odds ratio (OR) ranged from 1.42 [29] to 6.3 [30]) for older individuals compared to their younger counterparts. No significant difference in the OR of no or less utilization of LVR services between the age groups was reported in three studies [20,27,28] based on multivariable analysis.

Regarding education, five studies [19,25,26,29,36] using multivariate analysis indicated that lower academic achievement could be a barrier factor to LVR service utilization. Specifically, four of the five studies [19,26,29,36] indicated that higher educational attainment significantly decreased the OR of no or less utilization of LVR services compared to those with lower educational attainment (OR ranged from 0.38 [19] to 0.73 [29]). Similarly, one study [25] indicated that the OR for unmet refractive error and presbyopia needs was significantly higher among illiterate individuals than among their literate counterparts (3.5 and 3.1, respectively). However, no statistically significant association was found between academic attainment and LVR service utilization in two studies [20,28] based on multivariate analysis.

Ten studies examined economic status as a barrier factor to LVR service utilization, with eight reporting an association between less privileged economic status and no or less utilization of LVR services and three reporting no association between them. Six of the eight studies found less privileged economic status to be the barrier factor to LVR service utilization using descriptive (five studies) and qualitative (one study) analyses. For example, Wubben et al. [21] investigated the barrier factors to obtaining access to eye care and/or glasses among 142 indigenous people (mean age: 57 ± 11 years) in the Philippines using a questionnaire survey and descriptive analysis. The questionnaire survey included the following question: Does the cost of reading glasses prevent you from obtaining glasses? The responses were as follows: greatly (46.2%), moderately (15.2%), slightly (34.1%), and not at all (4.5%). In contrast, a study in China [20], with a total of 1008 study participants (mean age: 58.4 ± 10.7 years) from a rural area, revealed that lack of money was not reported as a common barrier factor to obtaining presbyopia correction (cf. only 1.2% of the respondents (n = 4/323) responded that lack of money was the most common barrier factor to obtaining presbyopia correction) based on a questionnaire survey and descriptive analysis.

Overall, eight of the 12 studies reported gender (female) as a non-barrier factor to LVR service utilization, with 6 using multivariate analysis. Specifically, gender (female) was unassociated with under-corrected or unconnected refractive errors [20,25,26,29], willingness to accept spectacle prescriptions [27], or unwillingness to pay for spectacles [30]. However, two studies [19,28] reported gender (female) as the barrier factor for under-corrected or unconnected refractive errors (OR:2.04) [19] and willingness to pay for eyeglasses (OR:0.39) [28].

Mixed findings were observed when the severity and type of eye conditions were examined as barrier factors to LVR service utilization. For example, cataracts were reported as a barrier factor (OR:1.5) for the state of under-corrected refractive error, according to a study from Singapore [36], with a study population of Chinese Singaporeans. However, a study [27] from China with a study population from rural provinces reported that postoperative presenting visual acuity, symptoms of hyperopia, and corneal astigmatism were not significantly associated with the willingness to accept spectacle prescriptions based on multivariate analysis.

Although they were the less commonly examined factors compared to the abovementioned five factors, “previous experience using eye care service (seven articles)” and “knowledge, information, and awareness (eight articles)” were also reported as the potential barrier factors to LVR service utilization. More than 80% of the studies examining these two factors provided supportive evidence that they could be barrier factors to LVR service utilization. The supportive evidence for “previous experience using eye care services (no)” and “knowledge, information, and awareness” was derived from mainly multivariate and descriptive analyses, respectively.

#### 3.3.2. Potential Barrier Factors from the Healthcare Settings and Society Categories

Five studies examined the potential barrier factors from healthcare settings and society categories as follows: “availability of an eye doctor (No)” and “recommendation from health center staff to attend the eye unit or vision center (No)” from the healthcare settings category, and “stigma, myth, or fear regarding eye care or assistive device” and “change of law about vision rehabilitation” from the society category. All five studies suggested that these factors could be barrier factors to LVR service utilization based on qualitative or descriptive analysis. Ormsby et al. [31], through in-depth interviews and qualitative analyses, suggested that some myths or fears surrounding seeking treatment (society category) could be a barrier factor and that recommendations for attending the eye unit or vision center from health care staff could lower the barrier factors to obtaining access to eye care service (healthcare settings category). Tanaka et al. [32] also reported that the change of law related to LVR services (society category) could extend the length of procedure for receiving the LVR service but could not influence the cost, frequency, and duration of LVR service utilization among Japanese individuals with vision impairments.

## 4. Discussion

### 4.1. Key Points of this Study’s Findings

To the best of our knowledge, this is the first scoping review to focus on LVR and the barrier factors to its utilization in ESEA. We reviewed 20 eligible articles and identified 21 potential barrier factors. Notably, age, education, economic status, “previous experience using eye care service”, and “knowledge, information, and awareness” were the possible barrier factors that were examined for their association with LVR service utilization in many eligible studies. These findings can enable stakeholders in the LVR sector to make LVR services more accessible for all individuals in need. However, this scoping review revealed research gaps in this study field. We found that the scope of the eligible studies was narrow. Additionally, the broadened facets of LVR and its barriers were not necessarily reflected in the eligible articles. The narrow research scope of the eligible articles made it difficult to extract comprehensive evidence on this topic. Therefore, further studies that address these research gaps are necessary to understand the whole picture of relevant evidence on LVR and the barriers to its utilization in ESEA.

### 4.2. What Are the Barrier Factors to LVR Service Utilization, and What Implications Can We Learn for Better LVR Service in ESEA

Age (older), education (lower), and economic status (less privileged) were possible barrier factors to LVR service utilization, followed by “previous experience using eye care services (no)” and “knowledge, information, and awareness”. This is the primary finding of this scoping review and the answer to the research question.

Our findings are consistent with those of previous reports that focused not only on ESEA countries and regions but also globally. Lam and Lest [38] conducted a systematic review to summarize the barrier factors to accessing LVR services. Although their findings were based on articles from Western countries, they reported “cost and income level”, “education level”, and “lack of awareness of low-vision care service” as possible barrier factors to accessing LVR service. A global survey of LVR service provision [9] with representatives from 195 countries also reported that age (older) and socioeconomic status (including financial issues) could be barrier factors to accessing LVR services. The World Health Organization [5], in their report titled World Report on Vision, overviewed the available evidence on barrier factors to eye care services and reported factors, including age (older), socioeconomic factors, and individuals’ perception of eye care services (including awareness, knowledge, and engagement) as potential barrier factors to eye care service utilization. Our findings imply that ESEA has common barrier factors to LVR service utilization as other countries and regions.

How the information extracted through this scoping review is used depends on whether the barrier factors are modifiable by interventions or programs or non-modifiable. Trenaman [39] argued that a non-modifiable contributing factor for a certain health-related outcome could be used to identify “who” to target (e.g., to identify systematically vulnerable or marginalized individuals for LVR service utilization). Non-modifiable factors include age, educational history, and economic status. These factors can be used to create a social support scheme targeting individuals with non-modifiable barrier factors (e.g., older people or less educated people) who have some systematic difficulty in accessing LVR services. Therefore, a program specifically focusing on vulnerable or marginalized individuals could be a promising solution to resolve the disparity in LVR service utilization. A modifiable contributing factor can be used to indicate “how” a certain health-related outcome can be improved (e.g., to formulate efficient and effective intervention programs to make the LVR service more accessible) [39]. Modifiable factors include awareness or knowledge of LVR services. A strategic public event, for example, aiming to raise awareness about eye conditions and LVR service, spread information on the availability or effectiveness of LVR service, or encourage local people to visit eye care professionals if they have problems with their eyes, can be a good example to use the information on modifiable barrier factors to formulate efficient and effective intervention programs aimed at making the LVR service more accessible.

Although gender disparity in eye health and access to eye care services has been well documented in previous reports [40,41,42], we found that gender, specifically female, was not reported as a barrier factor in many of the eligible articles. This finding implies that gender might be a less significant barrier factor to LVR service utilization in ESEA. Although the reasons for this discrepancy are unclear, one possible explanation is that our findings were mainly derived from studies that focused on LVR services related to spectacles or contact lenses. Generally, as previous reports have indicated [40,41,42], eye health status and access to eye care services may be disproportionately biased by gender. However, if we only focus on the utilization of LVR services related to spectacles or contact lenses, gender differences might not be a significant barrier factor. Therefore, careful considerations are required if LVR stakeholders in the ESEA regions apply this finding to accessibility to LVR services other than spectacle- or contact lens-related LVR services.

### 4.3. Research Gaps for Future Studies

This scoping review identified several research gaps that should be addressed in future studies to gain a comprehensive understanding of this topic.

First, the scope of the research should be expanded geographically. Approximately half of the eligible studies were conducted in China or Singapore. We identified only a few studies from countries and regions other than these two countries. Surprisingly, we identified only two eligible studies from Japan; nevertheless, one of the electronic databases used for searching for eligible articles was Ichushi-Web (Japanese medical literature database).

Second, the scope of LVR services and their barrier factors were narrow in the eligible articles. Specifically, the LVR services examined were primarily spectacles and contact lenses for individuals with refractive errors. The barrier factors examined were mainly from the individual category. A wider variety of LVR services (e.g., orientation and mobility training, braille training, or guide dog), clinical conditions (e.g., blind or narrowed visual fields), and barrier factors from the healthcare setting and Society category should be included, and their relationship with LVR service utilization should be examined in future studies. Moreover, given that ESEA is an ethnically diverse region, the factor of race or ethnicity, with only one eligible study identified in this scoping review, might be a prioritized barrier factor to be examined.

These research gaps can provide fragmented information and make it difficult to grasp the entire picture of the topic in ESEA. Therefore, further research is needed to bridge these gaps.

### 4.4. Study Limitation

This study had some limitations that should be considered. First, we might have missed potentially relevant articles because of language barriers and the limitations regarding the publication type included. This scoping review did not include articles written in languages other than English or Japanese. We also excluded publications other than original peer-reviewed and academic-quality articles, specifically unpublished studies, gray literature, and dissertations. Second, we operationally defined the LVR service for this scoping review as mentioned above and judged articles as eligible if clear descriptions consistent with our definitions were found in their articles. However, in some cases, only ambiguous descriptions (e.g., eye care or eye health services) were found in their articles. Additionally, we did not consider articles with ambiguous descriptions to be eligible, and this may have affected the selection process. Third, we only focused on individuals aged >50 years in ESEA. Therefore, our findings cannot be generalized to other generations, such as school children or adolescents. Finally, as mentioned earlier, non-barrier factors to LVR service do not necessarily mean that the factors are “enablers” for LVR service utilization. Further studies setting outcome variables as situations where the LVR service utilization was enhanced or accelerated are necessary for understanding the “enablers” for LVR service utilization. These limitations should be acknowledged when interpreting this study’s findings.

## 5. Conclusions

This scoping review identified 21 potential barrier factors to LVR service utilization in ESEA. The possible barrier factors, with supportive evidence in many eligible articles, were age, education, economic status, “previous experience using eye care services”, and “knowledge, information, and awareness”. Therefore, these findings can be used to make LVR services more accessible to people in ESEA. With further research addressing the research gaps identified through this scoping review, as well as further commitment among LVR service stakeholders, LVR services would be more accessible for all individuals in need in ESEA.

## Figures and Tables

**Figure 1 ijerph-20-07141-f001:**
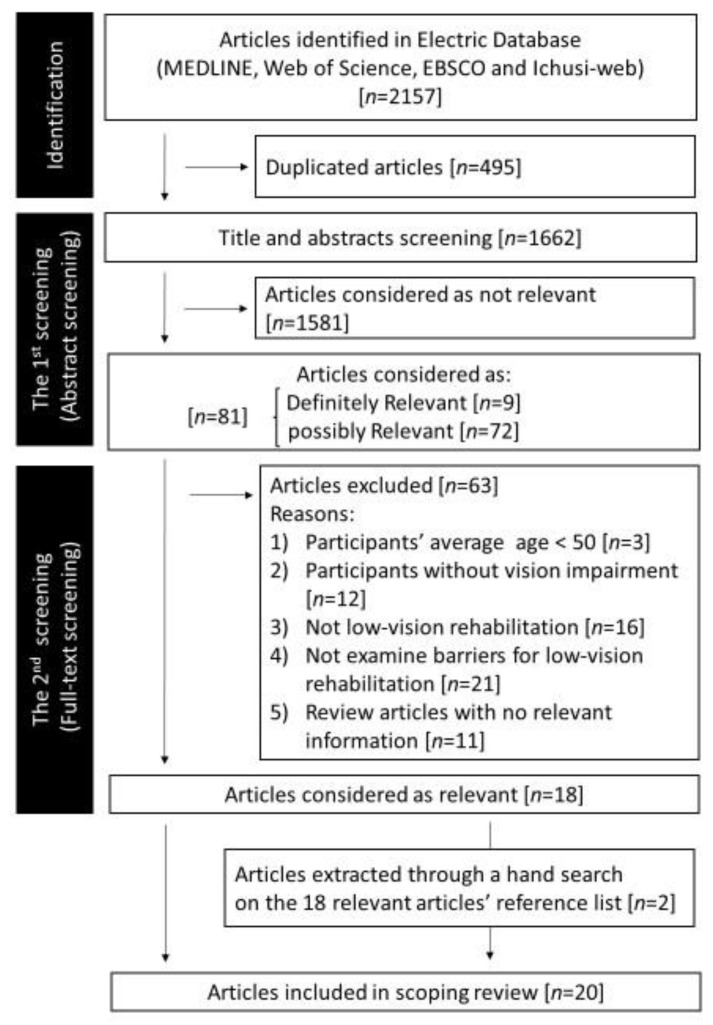
Flow diagram of the study selection. An electronic literature search of four electronic literature databases was performed on 18 and 19 April 2023.

**Table 1 ijerph-20-07141-t001:** Characteristics of included articles.

		*n* (%) ^†^
Year of publication		
2000–2009		6 (30)
2010–2019		11 (55)
2020–2023		3 (15)
Country		
China		7 (35)
Singapore		4 (20)
Cambodia		2 (10)
Japan		2 (10)
Timor-Leste		2 (10)
Indonesia		1 (5)
Taiwan		1 (5)
Philippines		1 (5)
Study methodology		
Quantitative study		17 (85)
Qualitative study		2 (10)
Mixed methods study		1 (5)
Study design		
Cross-sectional surveys		15 (75)
Prospective observational(cohort) studies		1 (5)
Qualitative studies		2 (10)
Mix methods research		1 (5)
Others		1 (5)
Type of location		
Rural		9 (45)
Urban		7 (35)
Urban and rural		2 (10)
Across country		1 (5)
Unclear		1 (5)
Types of study population		
General population		10 (50)
Individuals with visual impairment		8 (40)
Indigenous or minority people		2 (10)

^†^ Number of articles and percentage of the total 20 articles.

**Table 2 ijerph-20-07141-t002:** Characteristics of participants and low-vision rehabilitation in the included articles.

First Author, Year (Country)	Number of Participants (Male, Female)	Eye Disease (Eye Condition)	Low-Vision Rehabilitation		
			Types of LVR	Definitions of No or Less Utilization of LVR Service (e.g., Specific Situation or State of No or Less Utilization of LVR Services or Specific Question Regarding Barrier Factor to Obtaining Access to LVR Services)	Categories of Barrier Factors Which Were Examined for Their Relationship with LVR Access
Heine et al., 2019 (China) [18]	8268 (4093, 4175)	Not specific (Refractive error: farsightedness and nearsightedness)	Spectacles	Having vision loss, but spectacles are not used.	Individual
Zhu et al., 2013 (China) [19]	4545 (1910, 2635)	Not specific (Refractive error: farsightedness)	Spectacles	Those who have a visual acuity worse than 20/40 in the better eye without correction and could achieve 20/40 or better in the better eye with correction but do not wear spectacles or achieve such correction with their present spectacles.	Individual
Lu et al., 2011 (China) [20]	1008 (404, 604)	Not specific (Refractive error: nearsightedness)	Spectacles	Those with near vision < 20/50 due to functional presbyopia who do not have near-vision corrective spectacles or whose spectacles did not improve vision.	Individual
Wubben et al., 2014 (Philippine) [21]	142 (35, 107)	Not specific (Refractive error: nearsightedness)	Spectacles	Questions about the cost of reading glasses and the availability of an eye doctor as barriers to obtaining reading glasses.	Individual Health care settings
Lin et al., 2021 (China) [22]	5284 (3201, 2083)	Not specific (Refractive error: farsightedness)	Spectacles	Those who do not achieve visual acuity < 6/12 with current spectacles or have any spectacles at all.	Individual
Cheng et al., 2016 (China) [23]	5158 (2299, 2859)	Not specific (Refractive error: nearsightedness)	Spectacles	Questions about the common barriers to accessing near visual impairment correction (including spectacles).	Individual
Ramke et al., 2007 (Timor-Leste) [24]	1414 (721, 693)	Not specific (Refractive error: farsightedness and nearsightedness)	Spectacles	Refractive error requiring correction: Individuals with distance vision worse than 6/18 in the better eye, who do improve to at least 6/18 with pinhole, will benefit from refractive error correction (spectacles). Presbyopia requiring correction: individuals aged > 40 years with binocular near vision of worse than N8, who have at least 6/18 distance vision with pinhole, will benefit from presbyopia correction (spectacles). Questions about willingness to wear spectacles to improve vision and willingness to pay for them.	Individual
Ramke et al., 2012 (Timor-Leste) [25]	2014 (1044, 970)	Not specific (Refractive error: farsightedness and nearsightedness)	Spectacles	Those who had uncorrected (including under-corrected) refractive errors that would improve with appropriate correction to at least 6/18 in the better eye. Those who had either under-corrected or uncorrected binocular near vision worse than N8 that would improve with correction to at least N8.	Individual
Kuang et al., 2007 (Taiwan) [26]	1361 (No information)	Not specific (Refractive error: farsightedness)	Spectacles	Correctable visual impairment: presenting visual acuity (naked eye if without spectacles and with distance eyeglasses if worn) in the better eye of <6/12 that improved to no impairment (≥6/12) after refractive correction.	Individual
Congdon et al., 2007 (China) [27]	239 (87, 152)	Not specific (Refractive error: farsightedness and nearsightedness)	Spectacles	Those with near or distance visual acuity improved by >2 lines in either eye and being offered near and/or distance spectacles prescriptions to be filled at a nearby optical shop but refusing the prescription for spectacles or surgery. Questions about reasons for the refusal.	Individual
Saw et al., 2004 (Singapore) [36]	1152 (526, 626)	Not specific (Refractive error: farsightedness)	Spectacles	Under-corrected refractive error: those with improvement of at least two lines of better eye visual acuity with best refractive corrections.	Individual
Rosman et al., 2009 (Singapore) [37]	503 (238, 265)	Not specific (Refractive error: farsightedness)	Spectacles	Under-corrected refractive error: those with an improvement of at least 0.2 logMAR (two lines equivalent) in the best-corrected visual acuity in the better eye compared with the presenting visual acuity.	Individual
Bani et al., 2012 (Indonesia) [28]	193 (100, 93)	Cataract (Refractive error: farsightedness and nearsightedness)	Spectacles	Questions about willingness to pay 7 USD for near, distance, or bifocal spectacles.	Individual
Pan et al., 2014 (Singapore) [29]	10,014 (4924, 5090)	Not specific (Refractive error: farsightedness)	Spectacles or contact lens	Under-corrected refractive error: those with an improvement of at least 0.2 logMAR (equivalent to two lines) in the best-corrected visual acuity compared with presenting visual acuity in the eye with better visual acuity when presenting visual acuity was worse than 20/40 in the better eye.	Individual
Ramke et al., 2008 (Cambodia) [30]	293 (112, 181)	Not specific (Refractive error: farsightedness and nearsightedness)	Spectacles	Questions about willingness to pay for spectacles.	Individual
Ormsby et al., 2016 (Cambodia) [31]	62 (24, 38)	Not specific (Refractive error: farsightedness and nearsightedness)	Spectacles	Open-ended questions about the style, fit, costs, and satisfaction with wearing the spectacles and whether they would recommend the refraction services to others.	Individual Healthcare settings Society
Tanaka et al., 2012 (Japan) [32]	134 (No information)	Not specific (Not specific)	Comprehensive vision rehabilitation	Length of procedure for receiving the LVR service, cost, frequency(/W), and duration of service uptake.	Society
Luo et al., 2023 (China) [33]	8 (4, 4)	Not specific (Not specific)	Assistive devices, including reading glasses	Open-ended questions about factors influencing an individual’s choice of assistive devices.	Individual Society
Sekine et al., 2022 (Japan) [34]	18 (3, 15)	Not specific (Not specific)	Information about “aids and equipment” and “benefits and money” for individuals with vision impairments	Open-ended questions about how they correct information on “special aids and equipment” and “benefits and money” for individuals with vision impairments.	Individual Healthcare settings
Boey and Warren. 2019 (Singapore) [35]	106 (64, 42)	Not specific (Not specific)	Occupational therapy low-vision rehabilitation	Questions about: Reasons for discontinuation of occupational therapy low-vision rehabilitation services. Reasons for declining occupational therapy low-vision rehabilitation services.	Individual

**Table 3 ijerph-20-07141-t003:** A number of articles examine the barrier factors ^†^ to the utilization of low-vision rehabilitation using different analysis methods and their findings ^††^.

#	Factors	Findings of the Analysis	Methods of Analysis				Number of Articles	
			Quantitative Analysis		Qualitative Analysis	Descriptive Analysis	Sub-Total by Findings *N* (%) ^†††^	Total
			Univariate Analysis	Multivariate Analysis				
Individual category								
1	Gender (female)	Barriers	2 [24,37]	2 [19,28]	0	1 [18]	5 (42)	12 [18,19,20,22,24,25,26,27,28,29,30,37]
		Non-barriers	2 [22,24]	6 [20,25,26,27,29,30]	0	0	8 (67)	
2	Age (older)	Barriers	1 [22]	6 [19,25,26,29,30,36]	0	1 [18]	8 (62)	13 [18,19,20,22,24,25,26,27,28,29,30,36,37]
		Non-barriers	2 [24,37]	3 [20,27,28]	0	0	5 (38)	
3	Education (Lower)	Barriers	1 [24]	5 [19,25,26,29,36]	0	1 [18]	7 (70)	10 [18,19,20,24,25,26,28,29,36,37]
		Non- barriers	1 [37]	2 [20,28]	0	0	3 (30)	
4	Economic status (Less privileged)	Barriers	0	2 [29,30]	1 [31]	5 [18,21,23,24,27]	8 (80)	10 [18,19,20,21,23,24,27,29,30,31]
		Non-barriers	0	2 [19,30]	0	1 [20]	3 (30)	
5	Severity or Type of eye condition	Barriers	1 [22]	7 [19,20,26,27,28,29,36]	0	0	8 (80)	10 [19,20,22,26,27,28,29,30,36,37]
		Non-barriers	1 [37]	5 [20,27,28,29,30]	0	0	6 (60)	
6	Previous experience using eye care services (No)	Barriers	2 [24,37]	4 [26,27,29,36]	0	0	6 (86)	7 [24,26,27,29,30,36,37]
		Non-barriers	0	2 [26,30]	0	0	2 (29)	
7	Knowledge, information, awareness	Barriers	1 [37]	0	2 [31,34]	5 [20,23,24,27,35]	8 (100)	8 [20,23,24,27,31,34,35,37]
		Non-barriers	1 [37]	0	0	1 [20]	2 (25)	
8	Occupation (Less privileged)	Barriers	2 [24,37]	2 [19,25]	0	0	4 (80)	5 [19,24,25,30,37]
		Non-barriers	0	1 [30]	0	0	1 (20)	
9	Living place (Rural)	Barriers	1 [24]	2 [25,30]	0	1 [18]	4 (100)	4 [18,24,25,30]
		Non-barriers	1 [24]	0	0	0	1 (25)	
10	Race	Barriers	0	1 [29]	0	0	1 (100)	1 [29]
		Non-barriers	0	0	0	0	0 (0)	
11	Country of birth	Barriers	0	0	0	0	0	1 [29]
		Non-barriers	0	1 [29]	0	0	1 (100)	
12	Marital status (Not married)	Barriers	1 [24]	0	0	0	1 (33)	3 [24,26,29]
		Non-barriers	1 [24]	2 [26,29]	0	0	3 (100)	
13	Comorbidity, lifestyle	Barriers	0	0	0	0	0 (0)	2 [26,29]
		Non-barriers	0	2 [26,29]	0	0	2 (100)	
14	Requiring supportive service	Barriers	0	0	0	0	0	1 [26]
		Non-barriers	0	1 [26]	0	0	1 (100)	
15	Reading habit	Barriers	0	0	0	0	0 (0)	1 [27]
		Non-barriers	0	1 [27]	0	0	1 (100)	
16	Driving status	Barriers	0	0	0	0	0 (0)	1 [36]
		Non-barriers	0	1 [36]	0	0	1 (100)	
17	Physical Environment	Barriers	0	0	1 [33]	0	1 (100)	1 [33]
		Non-barriers	0	0	0	0	0 (0)	
Healthcare settings category								
18	Availability of an eye doctor (No)	Barriers	0	0	0	1 [21]	1 (100)	1 [21]
		Non-barriers	0	0	0	0	0 (0)	
19	Recommendation from health center staff to attend the eye unit or vision center (No)	Barriers	0	0	1 [31]	0	1 (100)	1 [31]
		Non-barriers	0	0	0	0	0 (0)	
Society category								
20	Stigma, myth, or fear regarding eye care or assistive device	Barriers	0	0	2 [31,33]	0	2 (100)	2 [31,33]
		Non-barriers	0	0	0	0	0 (0)	
21	Change of law about vision rehabilitation	Barriers	0	0	0	1 [32]	1 (100)	1 [32]
		Non-barriers	0	0	0	1 [32]	1 (100)	

^†^ The barrier factors mean that the factors are relevant to “less or no LVR service utilization”. While the non-barrier factors mean that the factors are not relevant to “less or no LVR service utilization”. Notably, non-barrier factors do not necessarily mean that the factors are “enablers” for LVR service utilization. ^††^ In some cases, more than one finding was extracted from one study because some studies included several analysis models. Findings extracted from one study can be contradictory (e.g., a study by Ramke et al. [24] reported gender (female] as both barrier and non-barrier factors based on different analysis models). ^†††^ Number of articles by findings and percentage of the total articles of each factor.

## Data Availability

No new data were created or analyzed in this study. Data sharing is not applicable to this study.

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
