# Peer review of "Barriers to the Utilization of Low-Vision Rehabilitation Services among Over-50-Year-Old People in East and Southeast Asian Regions: A Scoping Review"

_ijerph, 2023, doi:10.3390/ijerph20237141_

Round 1

Reviewer 1 Report

Comments and Suggestions for Authors

Manuscript ID:

Barriers to the utilization of low-vision rehabilitation services among over-50-year-old people in East and Southeast Asian regions: a scoping review

Authors: Saito Takashi and Imahashi Kumiko

Introduction

Topic of article is interest, because low-vision rehabilitation (LVR) is a special rehabilitation service for individuals with vision impairment that can help in improving their eye health status.

Material and methods

The PRISMA format is appropriately used in the article.

Eligibility criteria

The relationships between the outcome variables and explanatory (any barrier factors) variables were examined using quantitative, qualitative, or descriptive analysis.

I recommend not to present quantitative and qualitative data together.

Study design

Data charting

Data were charted by a single reviewer (ST) and another reviewer (IK) independently verified the extracted data - very detailed description of their activities.

Data synthesis contain three categories: As individual, healthcare setting and society

Results

Two reviewers (TS and IK) discussed the eligibility of the articles. Are the reviewers (TS and IK) the authors of this article?

Table 2: I would divide the data by location (China, Japan, Cambodia and other) or by LVR services utilization.

Table 3: I think there are a lot of factors being looked into in studies.

Discussion

The authors reviewed 20 eligible articles and identified 21 potential barrier factors. In particular, age, education, economic status, "previous experience using eye care" and "knowledge, information and awareness" were possible barrier factors.

Conclusions

This scoping review identified 21 potential barrier factors to LVR service utilization 446 in ESEA.

The presented work presents a high professional level.

Author Response

Thank you very much for your review and for your helpful comments.
I have attached our response to your comments.

Reviewer 2 Report

Comments and Suggestions for Authors

Thank you for this interesting paper. I have some comments which I would like you to consider and I have attached these for you.

Author Response

(The authors gave the same response as above.)

Reviewer 3 Report

Comments and Suggestions for Authors

This is a very thorough literature review. However, the discussion of the reviewed studies is quite brief. The actual discussion starts on page 12 (section 3.3.1) and, including conclusion section, is just over three and a half pages long, in an 18-page article.  There is surely room for a more extensive discussion of the findings of the articles. 

Author Response

(The authors gave the same response as above.)

Round 2

Reviewer 3 Report

Comments and Suggestions for Authors

Thank you for making the requested changes